# Nutrient Combinations Sensed by L-Cell Receptors Potentiate GLP-1 Secretion

**DOI:** 10.3390/ijms25021087

**Published:** 2024-01-16

**Authors:** Nalini Sodum, Orvokki Mattila, Ravikant Sharma, Remi Kamakura, Vesa-Pekka Lehto, Jaroslaw Walkowiak, Karl-Heinz Herzig, Ghulam Shere Raza

**Affiliations:** 1Research Unit of Biomedicine and Internal Medicine, Biocentre of Oulu, Medical Research Center, University of Oulu, Oulu University Hospital, Aapistie 5, 90220 Oulu, Finland; nalini.sodum@oulu.fi (N.S.); orvokki.mattila@oulu.fi (O.M.); sharma.ravikant@oulu.fi (R.S.); karl-heinz.herzig@oulu.fi (K.-H.H.); 2Department of Technical Physics, Faculty of Science, Forestry and Technology, University of Eastern Finland, 70210 Kuopio, Finland; vesa-pekka.lehto@uef.fi; 3Department of Gastroenterology and Metabolism, Poznan University of Medical Sciences, 60572 Poznań, Poland; jarwalk@ump.edu.pl

**Keywords:** nutrients, GLP-1, fatty acid, amino acid, enteroendocrine cells, appetite, obesity

## Abstract

Obesity is a risk factor for cardiometabolic diseases. Nutrients stimulate GLP-1 release; however, GLP-1 has a short half-life (<2 min), and only <10–15% reaches the systemic circulation. Human L-cells are localized in the distal ileum and colon, while most nutrients are absorbed in the proximal intestine. We hypothesized that combinations of amino acids and fatty acids potentiate GLP-1 release via different L-cell receptors. GLP-1 secretion was studied in the mouse enteroendocrine STC-1 cells. Cells were pre-incubated with buffer for 1 h and treated with nutrients: alpha-linolenic acid (αLA), phenylalanine (Phe), tryptophan (Trp), and their combinations αLA+Phe and αLA+Trp with dipeptidyl peptidase-4 (DPP4) inhibitor. After 1 h GLP-1 in supernatants was measured and cell lysates taken for qPCR. αLA (12.5 µM) significantly stimulated GLP-1 secretion compared with the control. Phe (6.25–25 mM) and Trp (2.5–10 mM) showed a clear dose response for GLP-1 secretion. The combination of αLA (6.25 µM) and either Phe (12.5 mM) or Trp (5 mM) significantly increased GLP-1 secretion compared with αLA, Phe, or Trp individually. The combination of αLA and Trp upregulated GPR120 expression and potentiated GLP-1 secretion. These nutrient combinations could be used in sustained-delivery formulations to the colon to prolong GLP-1 release for diminishing appetite and preventing obesity.

## 1. Introduction

Obesity is a multifactorial chronic disease with excess adiposity [1]. It is a well-known risk factor for metabolic and cardiovascular diseases (CVDs) [2]. Diet and physical activity are two of the major regulators of obesity. Recently, the USFDA approved glucagon-like peptide-1 (GLP-1) agonists for the treatment of obesity. GLP-1 is secreted from the enteroendocrine L-cells of the intestine after ingestion of food. L-cells are found throughout the gastrointestinal (GI) tract, originating in the proximal small intestine and gradually increasing in density towards the distal region of the gut [3]. GLP-1 suppresses food intake, delays gastric emptying, and stimulates insulin secretion from the pancreas [4]. Therefore, GLP-1 has gained significant attention as a therapeutic agent for diabetes and obesity. GLP-1 (7–36) and GLP-1 (7–37) amides are the active forms in human plasma, and are quickly degraded by the enzyme dipeptidyl peptidase-4 (DPP4), leading to inactive GLP-1 (9–36) and GLP-1 (9–37) amides [5]. Only 10–15% of the secreted GLP-1 reaches the systemic circulation in active form and the half-life of GLP-1 is approximately 1–2 min. Hence, employing GLP-1 as a pharmaceutical compound in its native form is not very suitable [6,7].

Nutrients such as carbohydrates, fats, and proteins stimulate GLP-1 secretions [5]. In animals and humans, unsaturated fatty acids are more potent GLP-1 secretagogues than saturated fatty acids [8,9]. Chain length and degree of FFA saturation influence GLP-1 secretion. Longer-chain-length FFAs stimulate gut peptide secretion more potently than shorter-chain FFAs [10,11]. In particular, long-chain polyunsaturated fatty acids (LCPUFAs) such as docosahexaenoic acid (DHA 22:6, n-3), α-linolenic acid (αLA, C18:3, n-3), and eicosapentaenoic acid (EPA, 20:5, n-3) have shown a higher potency than other LCFAs in suppression of body weight gain and appetite [12,13]. The FFAs bind to several receptors on the surface of intestinal L-cells, including G protein-coupled receptors (GPCRs). In addition, other digestion products such as glucose and amino acids also act as stimuli for L-cells via different receptors such as calcium signaling receptor (CaSR), ion channels and transporters (peptide transporter-1 (PEPT1), and sodium/glucose co-transporter 1 (SGLT1) [14,15]. Peptides and amino acids stimulate both cholecystokinin (CCK) and GLP-1 release from the enteroendocrine cell line STC-1 [16].

GLP-1 secretion signals the presence of food in the intestine and suppresses food intake centrally via the vagal nerve [17,18]. GLP-1 exerts its actions through the glucagon-like peptide-1 receptor (GLP1R), which is expressed in numerous tissues including the pancreas, kidney, heart, lung, adipose, smooth muscle, and nerve cells [19]. Fasting GLP-1 concentration in humans typically ranges from 5 to 15 pmol/L and increases 2–4-fold after food ingestion [3]. In insulin resistance, GLP-1 secretion is impaired [20,21,22,23]. In addition, several studies in humans have shown that GLP-1 secretion is impaired in obese subjects [24,25,26]. In contrast, a meta-analysis of 22 trials showed that there was no difference in glucose-stimulated GLP-1 secretion between type-2 diabetics and healthy controls [27]. Recently, a study reported that diabetic subjects had higher fasting and glucose-stimulated GLP-1 levels compared with prediabetic and normal subjects [28]. This inconsistent finding could be due to diagnostic criteria for prediabetes and diabetes, sample size, or detection method for GLP-1 and GLP-1 measurement timing [21,22]. Bariatric surgery is used to treat morbid obesity, resulting in a decrease in food intake and an increase in endogenous postprandial gut hormone secretions, particularly GLP-1 [29,30,31]. Bariatric surgery also demonstrates that increased distal nutrient delivery has the potential to activate a greater number of GLP-1-producing enteroendocrine cells and to enhance endogenous GLP-1 secretion [29].

Several chimeric peptides have been developed to combine their beneficial physiological response [32]. Pan et al. created a novel hybrid peptide utilizing the physiological actions of GLP-1 and glucagon in glucose homeostasis [33]. In addition, dual GLP-1 receptor and glucagon receptor agonist reversed obesity in high-fat-diet-fed mice by daily injections or by employing a polyethylene glycolated agonist injected once per week [34,35]. The synthetic peptide, SAR441255, is a novel tri-agonist of the GLP-1, glucagon, and GIP receptors with better glycemic control in healthy subjects [36]. These studies have demonstrated that hybrid peptides are a suitable approach to combining the physiological functions of different peptides to treat the disease [37,38]. Rather than exogenous application of the various drugs, we wanted to stimulate endogenous GLP-1 secretion with nutrient combinations.

We hypothesized that the combination of different nutrients such as FFAs and amino acids stimulates additively or potentiates endogenous GLP-1 secretion via the different receptors on L-cells. Therefore, we investigated the combined effects of alpha-linolenic acid (αLA) and the amino acids L-phenylalanine (Phe) or L-tryptophan (Trp) for GLP-1 secretion using the enteroendocrine cell line STC-1.

## 2. Results

### 2.1. In Vitro GLP-1 Secretions in STC-1 Cells

αLA (6.25–12.5 µM), Phe (6.25–25 mM), and Trp (2.5–10 mM) dose-dependently stimulated GLP-1 secretion in STC-1 cells (Figure 1). αLA (12.5 µM) significantly stimulated GLP-1 secretion ~5-fold and αLA (6.25 µM) ~4-fold, compared with the buffer (control). Similarly, Trp (2.5–10 mM) stimulated GLP-1 secretion ~2–3.5-fold compared with the control. Interestingly, the combination of αLA (6.25 µM) and Trp (5 mM) significantly (~2.5-fold) increased GLP-1 secretion compared with either αLA (6.25 µM) or Trp (5 mM) individually. In addition, the combination of αLA (6.25 µM) and Trp (5 mM) significantly increased GLP-1 secretion compared with a higher dose of αLA (12.5 µM). Higher-dose combinations of αLA (12.5 µM) and Trp (10 mM) and lower-dose combinations αLA (3.125 µM) and Trp (2.5 mM) increased GLP-1, but not statistically significantly, compared with individual nutrients (Figure 1).

Similarly, Phe (6.25–25 mM) showed a clear dose response in GLP-1 secretion compared with the buffer (control). The combination of Phe (12.5 mM) and αLA (6.25 µM) increased GLP-1 secretion ~3-fold compared with αLA (6.25 µM) and Phe (12.5 mM) alone (Figure 2). In addition, the combination of Phe (12.5 mM) and αLA (6.25 µM) significantly increased GLP-1 secretion compared with the higher dose of αLA (12.5 µM). The higher dose combinations of αLA (12.5 µM) and Phe (25 mM) and the lower dose combination of αLA (3.125 µM) and Phe (6.25 mM) increased GLP-1 secretion, but it was not significant compared with individual nutrients (Figure 2).

### 2.2. Gene Expression

#### mRNA Expression Levels of Fatty Acid and Amino Acid Receptors

αLA, Phe, and Trp, and their combinations, increased mRNA gene expression of GPR120 and GPR40 compared with the control (Figure 3).

The combination of αLA (6.25 µM) and Trp (5 mM) upregulated GPR120 mRNA expression compared with the positive control αLA. No significant change in GPR119 or GPR40 was found with αLA, Phe, or Trp, or the combinations of αLA and Trp or αLA and Phe (Figure 3). The nutrients Phe and Trp, and their combinations with αLA, showed an upregulating trend in CaSR and GPRC6A mRNA expression compared with the controls (Figure 4).

## 3. Discussion

Fatty acids such as αLA and L-amino acids such as Phe and Trp significantly increased GLP-1 secretions compared with the control (Figure 1 and Figure 2). Our results are consistent with previous findings on GLP-1 secretion with fatty acids and amino acids in STC-1 cells [39,40,41]. In STZ-induced diabetic rats, oral administration of linoleic acid increased GLP-1 secretion and reduced postprandial hyperglycemia [42]. Adachi et al. demonstrated that unsaturated long-chain fatty acid infusion into the colon of mice increased GLP-1 secretion, but saturated and middle-chain fatty acids did not [43]. Oral ingestion of corn oil (99% triacylglycerol with 59% polyunsaturated FA, 24% monounsaturated FA, and 13% saturated FA) induced GLP-1 secretion in wild-type mice and in GPR120^−/−^ mice, but not in GPR40^−/−^ mice [44]. The author demonstrated that partial GPR40 agonists did not stimulate GLP-1 secretion but the combination of partial and full agonists cooperatively enhanced receptor activation and GLP-1 secretion both in vitro and in vivo [44]. L-Phe stimulated GLP-1 secretion in both STC-1 cells and murine primary colonic L-cell cultures [41]. The author found that L-Phe stimulated GLP-1 secretion in STC-1 cells via CaSR [45]. Oral administration of L-Phe in rodents increased plasma GLP-1 levels and reduced food intake and body weight. CaSR inhibition attenuated the anorectic effect of intra-ileal L-Phe in rats and GLP-1 release from STC-1 and primary L-cells [41]. These results suggest that the effect of Phe on GLP-1 secretion is mediated via CaSR. In humans, intragastric administration of 10 g L-Phe in healthy males suppressed energy intake and increased plasma CCK, but did not affect gastric emptying and plasma GLP-1 levels [46]. The authors reported a reduction in plasma glucose, which they did not explain. Surprisingly, they did not add DPP4 inhibitors for GLP-1 protection, which might have contributed to their GLP-1 results [46]. Oral ingestion of 10 g L-Phe increased plasma GIP but did not change plasma GLP-1, energy intake, or appetite with no DPP4 inhibitors added [47]. L-Phe is primarily absorbed in the proximal gut, while D-Phe reaches the distal gut [48]. Various L-amino acids such as L-Phe, L-Trp, L-proline, L-arginine (Arg), and L-glutamic acid increased GLP-1 secretion in STC-1 cells [49]. The author reported that L-Phe was the most potent among the amino acid in terms of GLP-1 secretion, activating GPR142 receptors [49].

In perfused rat intestine, Modvig and colleagues found that different amino acids such as L-valine, L-Trp, and L-Arg at a concentration of 50 mM stimulated GLP-1 secretion [50]. GLP-1 secretion was increased after intraduodenal administration of L-Trp (100 mg/kg body weight) in rats [40]. In contrast, intraduodenal infusions of L-Trp, leucine, Phe, and glutamine in humans resulted in reduced food intake but no significant differences in plasma GLP-1 levels were found [51]. Again, no DPP4 inhibitor was added for protection of GLP-1 [51]. In obese participants, L-Trp (1.56 g orally) increased GLP-1 secretion and reduced gastric emptying; however, L-Leucine did not stimulate GLP-1 secretion [52]. Furthermore, in humans, oral administration of an amino acid mixture containing 3 g L-arginine, 3 g L-Trp, 6 g L-glutamine, and 6 g L-leucine, modestly reduced appetite and increased GLP-1 levels in obese adolescents [53]. In healthy, lean men, whey protein containing all amino acids including L-Trp increased the GLP-1 and insulin [54].

Our results showed that the combination of a fatty acid with amino acids potentiated GLP-1 secretion in STC-1 cells (Figure 1 and Figure 2). In humans the active GLP-1 is very low (<2 pmol/L) in the fasting state, and peaks around 5–10 pmol/L (~2–5-fold increase) after nutrient consumption [55,56]. We measured active GLP-1 from L-cells. We used an established cell line as an in vitro model, in which individual nutrients increased GLP-1 secretion ~2–3.5-fold compared with the control. However, combinations of nutrients αLA (6.25 µM) and Trp (5 mM), and αLA (6.25 µM) and Phe (12.5 mM) increased GLP-1 secretions ~10-fold compared with the control. In mice, a mixed meal induced more GLP-1 secretions than isocaloric glucose [57], indicating that the presence of all macronutrients may synergistically enhance GLP-1 secretion. In healthy men, the combined intraduodenal administration of 5.55 g lauric acid and 4.07 g L-Trp substantially reduced energy intake with a marked stimulation of CCK and suppression of ghrelin [58]. In addition, the author showed that only lauric acid raised plasma GLP-1, and not the combination of lauric acid and Trp, indicating that the effect on GLP-1 was due to lauric acid. We found an increase in GLP-1 secretion with the αLA and L-Trp combination. The synergistic effects could be due to the activation of different receptors in STC-1 cells. Free fatty acids are sensed mainly via GPR120 and GPR40, which are present on the L-cells [39]. αLA is well known to stimulate GPR120 and GPR40 receptors and increase GLP-1. GPR40 is a fatty acid receptor with a high affinity for medium- to long-chain saturated and unsaturated fatty acids [59]. Both GPR120 and GPR40 receptors enhance intracellular Ca^2+^ levels [60]. We found that αLA upregulated GPR120 and GPR40 mRNA expression, but this was not significant. αLA (6.25 µM) and the combination αLA 6.25 µM and Trp 5 mM showed similar GPR40 mRNA expression, which is mediated via fatty acid αLA. However, we found that the combination of αLA and L-Trp upregulated GPR120 expression (Figure 3). Both L-Phe and L-Trp activate CaSR and stimulated GLP-1 secretion in STC-1 cells and in rodents [40,41,45,50]. Our results showed an upregulating trend in CaSR expression with L-Phe and L-Trp in STC-1 cells; however, fatty acid αLA and the control showed similar CaSR expression. These results indicate that the combination of αLA and amino acid (Trp/Phe) stimulates their respective receptors and potentiates GLP-1 secretion in our cell model (Figure 4). In addition, it has been shown that L-Phe and L-Trp were sensed by the GPR142 receptor [61] and L-Trp is an agonist of GPR142, stimulating GLP-1 secretion [62]. In addition, we found that the lower-dose combination of αLA 6.25 µM and Trp 5 mM showed higher GPR120, GPR40, GPR119, and GPRC6A mRNA expression than the higher-dose combination of αLA (12.5 µM) and Trp (10 mM). The reduced GLP-1 secretion with a higher dose combination of αLA and amino acids Trp/Phe might have been due to desensitization of receptor signaling. The higher and lower dose combinations showed similar expressions of the above receptors. Amino acids and dietary peptides elicit their effects via GPCRs including CaSR, GPR35, GPR93, GPR142, and GPRC6A, and the umami taste receptor (Tas1R1/Tas1R3), stimulating the release of GLP-1 and PYY [50]. These results indicate that the combination of fatty acids and amino acids is sensed via different receptors and enhances GLP-1 secretion. Our results will need now to be tested in animals and humans for further confirmation.

## 4. Materials and Methods

### 4.1. Materials

Dulbecco’s modified eagle medium (DMEM, Cat. No. P04-03500, Aidenbach, Germany), α-linolenic acid (Cat. No. L2376), DPP4 inhibitor (Cat. No. DPP4-010), and active GLP-1 ELISA kit (Cat. No. EGLP-35K) were purchased from Merck (Darmstadt, Germany). Other materials were horse serum Gibco (Cat. No. 16050-122, Thermo Fisher, Waltham, MA, USA), fetal bovine serum Gibco (Cat. No. 10270106, Fisher Scientific, Waltham, MA, USA), L-glutamine Gibco (Cat. No. 25030-024, Thermo Fisher Scientific, Waltham, MA, USA), penicillin–streptomycin Gibco (Cat. No. 15140-122, Thermo Fisher, Waltham, MA, USA), L-Phe (P5482-Merck), L-Trp (T0254-Merck), T-75 flasks (Sarstedt, AG & Co. KG, Nümbrecht, Germany), Nunc Cell-Culture Treated 24-well multi-dishes (Thermo Fisher Scientific, Waltham, MA, USA), RNA NucleoSpin kit (Macherey Nagel GmbH & Co. KG, Düren, Germany), SensiFast cDNA synthesis kit (Meridian Biosciences Inc., Cincinnati, OH, USA), and PowerUp™ and SYBR™ PowerUp™ SYBR™ Green Master Mix for qPCR (A25743, Applied Biosystem, Waltham, MA, USA). Mouse-specific primers for Rt-PCR were purchased TAG Copenhagen, (Copenhagen, Denmark).

### 4.2. STC-1 Cell Culture

The STC-1 cell line was cultured in T-75 flasks in DMEM supplemented with 15% horse serum, 2.5% fetal bovine serum (FBS), 1% L-glutamine, and penicillin–streptomycin, as previously described [63,64]. Cells were maintained in the above media at 37 °C and 5% CO_2_, and passage numbers 14–22 were used for GLP-1 secretions.

### 4.3. In Vitro GLP-1 Secretions from STC-1 Cells

STC-1 cells were seeded at 1.0 × 10^5^ cells/500µL media per well in 24-well cell culture plates. Cells were counted by an automated cell counter (LUNA-Ⅱ™, Logos Biosystems, Inc., Villeneuve-d’Ascq, France). Cells were incubated until they reached 85–90% confluent. On the day of experiments, cells were washed twice with Krebs-Ringer Bicarbonate Buffer (Krebs; 118 mM NaCl, 4.7 mM KCl, 25 mM NaHCO_3_, 1.25 mM CaCl_2_, 1.2 mM MgSO_4_, and 1.2 mM KH_2_PO_4_, pH to 7.4), and incubated with buffer for 1 h. After 1 h of preincubation, the buffer was aspirated, and cells were treated with different nutrients and combinations of αLA, Phe, and Trp along with DPP4 inhibitor in Krebs buffer for 1 h at 37 °C: αLA (12.5 µM and 6.25 µM), Phe (6.25–25 mM), and Trp (2.5–10 mM) and their combinations: αLA + Phe (i. 12.5 µM + 25 mM, ii. 6.25 µM + 12.5 mM, and iii. 3.125 µM + 6.25 mM) and αLA + Trp (i. 12.5 µM + 10 mM, ii. 6.25 µM + 5 mM, and iii. 3.125 µM + 2.5 mM) were used. Krebs buffer with ethanol (0.1%) and DPP4 inhibitor (0.25%) were used as control. After 1 h, supernatants were collected, centrifuged at 12,000 rpm for 5 min at 4 °C and stored at −70 °C. Active GLP-1 was measured from the collected supernatant by GLP-1 ELISA kit as per manufacturer instructions [39]. Cells were harvested for gene expression analysis by real-time PCR.

### 4.4. Gene Expression (RT-PCR)

Gene expression analysis was performed by real-time PCR. RNA was extracted from cell lysates with the total RNA NucleoSpin kit. Genomic DNA was removed by gDNA columns, and concentrations and quality of the RNA were measured by NanoDrop ND-1000 ultraviolet–visible (UV–vis) spectrophotometer (NanoDrop Technologies, Wilmington, DE, USA). cDNAs were synthesized from 1 μg of RNA using SensiFast cDNA synthesis kit as per manufacturer instructions. Mouse-specific primers for fatty acid receptors for GPR120 (Far: GGCACTGCTGGCTTTCATA; Rev: GATTTCTCCTATGCGGTTGG), GPR40 (Far: TTTCATAAACCCGGACCTAGGA; Rev: CCAGTGACCAGTGGGTTGAGT), and GPR119 (Far: CTT GCT GTC CTA ACC ATC CTC A; Rev: CCA CGC CAA TCA AGG TAT CAG), and amino acid receptors for GPRC6A (Far: CCC AGT CTT GTC ATA CCC CAG; Rev: TGC TGT GTA TCA TAG CCA GAG T) and CaSR (Far: AGC AGG TGA CCT TCG ATG AGT; Rev: ACT TCC TTG AAC ACA ATG GAG C) were used. Samples were normalized to glyceraldehyde 3-phosphate dehydrogenase (GAPDH) as endogenous control. PCR reactions were performed in QuantStudio 5 (ThermoFisher Scientific, Waltham, MA, USA) in a total volume of 20–30 μL. All samples were measured in duplicate using the following conditions: 2 min at 50 °C and 10 min at 95 °C, followed by 42 cycles of 15 s at 95 °C and 1 min at 62 °C. Each assay included a standard curve of three serial dilutions of cDNA from the fasted mouse and no template controls. Results were calculated according to the instructions of the manufacturer (ThermoFisher Scientific, Waltham, MA, USA). Gene expression analysis was conducted by delta delta ct method.

### 4.5. Statistical Analysis

One-way analysis of variance (ANOVA) was used to analyze for statistical significance between the groups using GraphPad Prism, version 7 (GraphPad Software, Inc., La Jolla, CA, USA) and Dunnett’s multiple comparison test was used to analyze the difference between treatment groups. Individual nutrients were compared with buffer, and nutrient combinations were compared with their respective nutrients alone. The values are represented as mean ± standard errors of mean (SEM) and differences were considered statistically significant when *p* < 0.05 and *n* = 6.

## 5. Conclusions

Fatty acids such as α-linolenic acid and amino acids phenylalanine or tryptophan stimulated GLP-1 secretion. The combination of α-linolenic acid with either tryptophan or phenylalanine potentiated GLP-1 secretion. However, these nutrients are absorbed in the proximal intestine and do not reach to the distal part of intestine where most of GLP-1-secreting cells are localized. The combination of these nutrients could be used in sustained-delivery formulations such as mesoporous particles [6] to the colon as food additives, increasing endogenous GLP-1 release, improving insulin secretion, and reducing appetite.

## Figures and Tables

**Figure 1 ijms-25-01087-f001:**
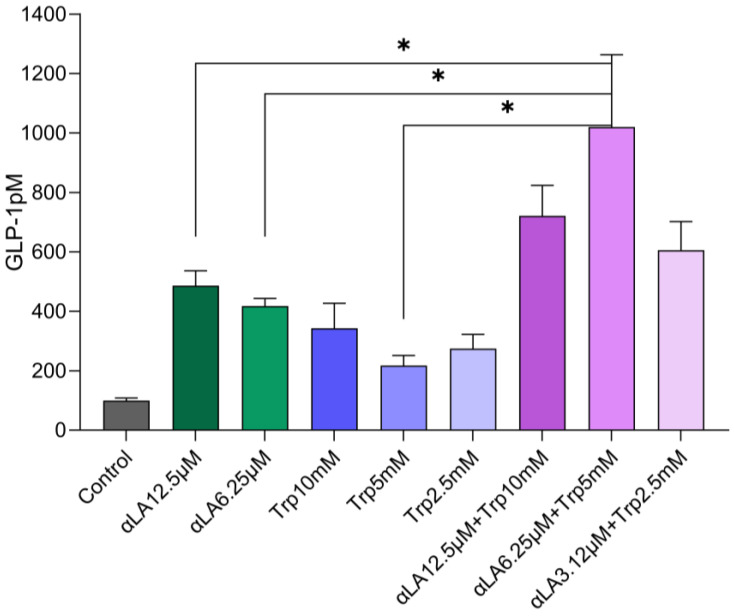
Alpha linolenic acid (αLA) (12.5 µM) significantly stimulated GLP-1 secretion compared with the control. The combination of αLA (6.25 µM) and L-tryptophan (Trp) (5 mM) increased GLP-1 secretion compared with individual nutrients. In addition, the combination of αLA (6.25 µM) and Trp (5 mM) significantly increased GLP-1 secretion compared with the higher dose of αLA (12.5 µM). The values represent the mean ± standard error of mean (SEM), and differences were considered statistically significant when *p* < 0.05 and *n* = 6 (* *p* < 0.05).

**Figure 2 ijms-25-01087-f002:**
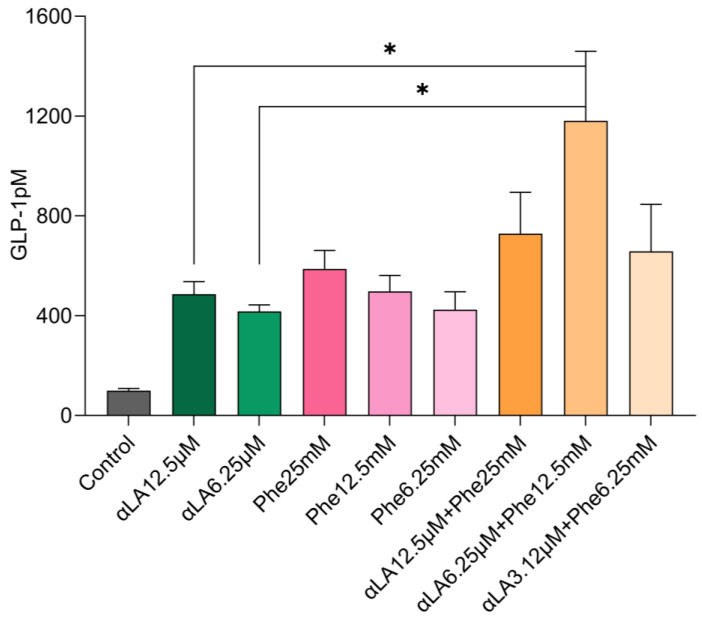
αLA (12.5 µM) significantly stimulated GLP-1 secretion compared with the control. The combination of αLA 6.25 µM and L-phenylalanine (Phe) (12.5 mM) increased GLP-1 secretion compared with individual nutrients. The values represent the mean ± standard error of mean (SEM), and differences were considered statistically significant when *p* < 0.05 and *n* = 6 (* *p* < 0.05).

**Figure 3 ijms-25-01087-f003:**
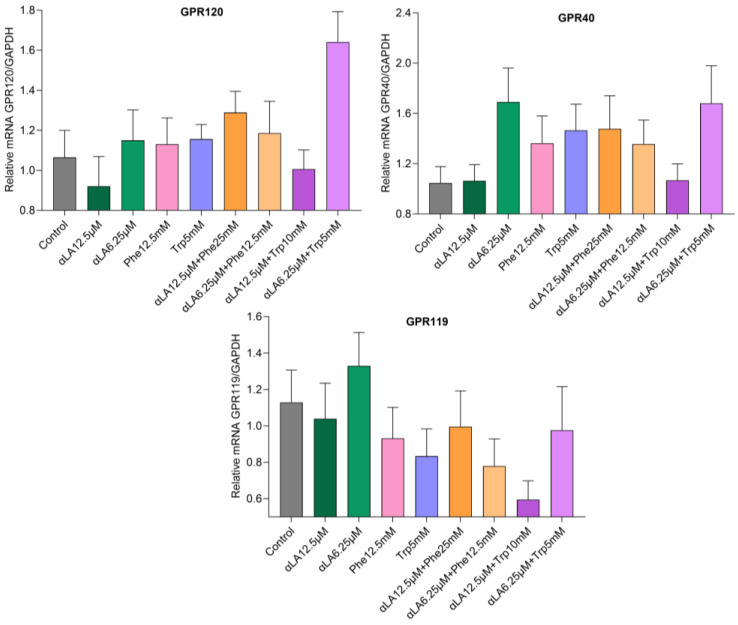
The combination of αLA (6.25 µM) and Trp (5 mM) upregulated GPR120 mRNA expression compared with the positive control αLA. There was no significant change in GPR119 or GPR40. The values represent the mean ± standard error of mean (SEM), and differences were considered statistically significant when *p* < 0.05 and *n* = 6.

**Figure 4 ijms-25-01087-f004:**
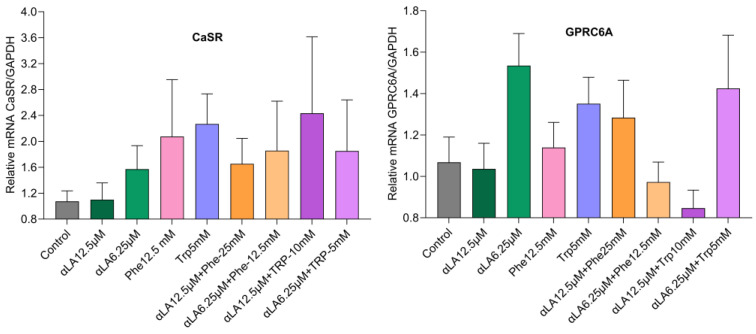
Phe and Trp and their combinations with αLA showed an upregulating trend in CaSR and GPRC6A mRNA expression compared with the control. The values represent the mean ± standard error of mean (SEM), and differences were considered statistically significant when *p* < 0.05 and *n* = 6.

## Data Availability

Data is contained within the article.

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
