# Peer review of "Nutrient Combinations Sensed by L-Cell Receptors Potentiate GLP-1 Secretion"

_ijms, 2024, doi:10.3390/ijms25021087_

Round 1
Reviewer 1 Report
Comments and Suggestions for Authors
I congratulate you for your excellent work.
As regards the methods, please define control more clearly.
I would suggest to expand the discussion regarding your results by commenting the observed differences between different concentrations of αLA, and Trp, and αLA, and Phe, and, may be, propose an explanation for these observed differences.
Also, the authors may comment in what extent the observed increase in GLP1 may translate in clinical significance in humans.
Comments on the Quality of English LanguageEnglish Language is fine, but I spotted minor editing issues, for example in row 83, 177, 196, 268.
Author Response
Response: We thank the reviewers for their valuable time and comments to improve our manuscript. We changed our manuscript accordingly. Please see our point-to-point responses:
- As regards the methods, please define control more clearly.
Response: The control is now defined and included in the manuscript (Line 99, Line 115; page 3 and line 259 page 8): “KREBS buffer with ethanol (0.1%) and DPP4 inhibitor (0.25%) were used as a control”. Comparisons between control and treatment groups were added in statistical analysis (line 290, page 8): Individual nutrients were compared with buffer and nutrient combinations were compared with their respective nutrients alone.
- I would suggest to expand the discussion regarding your results by commenting the observed differences between different concentrations of αLA, and Trp, and αLA, and Phe, and, may be, propose an explanation for these observed differences.
Response: Thank you very much for your suggestion. We now expanded the discussion on Page 7 (line 207) “αLA 6.25 µM and combination αLA 6.25 µM + Trp 5 mM has shown similar GPR40 mRNA expression which is mediated via fatty acid αLA. (Line 212) Our results showed an upregulating trend in CaSR expressions with L-Phe and L-Trp in STC-1 cells, however fatty acid αLA and control show similar CaSR expression. These indicate that the combinations of αLA with amino acid (Trp/Phe) stimulate their respective receptor and potentiates GLP-1 secretion. (Line 217) In addition, we found that lower dose combination of αLA 6.25 µM + Trp 5 mM showed higher GPR120, GPR40, GPR119, and GPRC6A mRNA expression than higher dose αLA12.5 µM + Trp 10 mM. The reduced GLP-1 secretion with a higher dose combination of αLA with amino acids Trp/Phe might be due to desensitization of receptor signalling. The higher and lower dose combinations showed similar expressions of the above receptors.
- Also, the authors may comment in what extent the observed increase in GLP1 may translate in clinical significance in humans.
Response: Thank you very much for your comment. We added clinical significance in discussion. In humans, blood concentrations of GLP-1 generally range between 5 pmol/L and 15 pmol/L in fasting state and increase two- to four-folds after food ingestion (Holst JJ.2007). (Line 188, Page 6) In humans the active GLP-1 is very low (< 2 pmol/l) in fasting state, which peaks around 5–10 pmol/l ( ̴ 2-5 times) after nutrient consumption (Kuhre, Rune Ehrenreich et al. 2015 and Holst, J J, and C F Deacon.2005). We measured active GLP-1 from L cells. We used an established cell line as in vitro model, in which individual nutrients increased GLP-1 secretion ̴ 2-3.5-fold compared to control. However, combinations of nutrients αLA 6.25µM+Trp5mM, and αLA 6.25µM+Phe12.5mM showed ̴ 10 folds higher GLP-1 secretions compared to control. (Line 225, page 7) Our results have been obtained in a well established in vitro system and would need now to be tested in animals and humans for further translation.
- Comments on the Quality of English Language: English Language is fine, but I spotted minor editing issues, for example in row 83, 177, 196, 268.
Response: Thank you very much for your kind suggestion, these errors are corrected throughout the manuscript. Row 83 (Now 86), Row 177 (Now181), Row 196 (Now 205), Row 268 (Now 289). In addition, the sentence has been modified (line 86; page 2): “Rather than exogenously application of the various drugs, we would like to stimulate endogenous GLP-1 secretion with nutrient combinations”.
Reviewer 2 Report
Comments and Suggestions for Authors
This paper certainly deserves publication as reported data seems relevant and interesting. However, several changes are necessary before it can be accepted. Manuscript addresses a relevant research theme as involvement of GLP-1 action and of its receptor activity is currently getting major attention for management of obesity and diabetes. Observed synergies between linolenic acid and amino acids are certainly impressive and could also be effective in pathophysiological conditions beside culture systems. Data from cell treatments with nutrients are clear and relevant about GLP-1 secretion, but are not adequately supported from expression assays, which should receive a deeper discussion and adequate interpretation. Otherwise, the association of the two set of data is not justified. Secretion of GLP-1 and expression of receptors, which are firstly to be clearly identified more than as GPR120, GPR40, etc., require a better understanding of current status of knowledge as well as possible explanations of their relationships within the experimental system. Reasons for the choice of each analyzed mRNA should also be provided to explain the experimental logic and data interpretation. Authors state that “monounsaturated FFA induced higher GLP-1 secretion than polyunsaturated or saturated FFA”. However, as polyunsaturated alpha-linolenic acid was used for their treatments, an explanation for this choice should be provided. Abstract reports that DPP4 inhibitor, still reported only as an undefined abbreviation, was used with nutrient combinations, but this is not described in results and methods. In addition, if DPP4 inhibitor was added to combinations, data should also include controls with it and without it.
Comments on the Quality of English LanguageMinor English corrections are necessary as text in several points is not fully clear.
Author Response
Response: We thank the reviewers for their valuable time and comments to improve our manuscript. We changed our manuscript accordingly. Please see our point-to-point responses:
This paper certainly deserves publication as reported data seems relevant and interesting. However, several changes are necessary before it can be accepted. Manuscript addresses a relevant research theme as involvement of GLP-1 action and of its receptor activity is currently getting major attention for management of obesity and diabetes. Observed synergies between linolenic acid and amino acids are certainly impressive and could also be effective in pathophysiological conditions beside culture systems.
- Data from cell treatments with nutrients are clear and relevant about GLP-1 secretion, but are not adequately supported from expression assays, which should receive a deeper discussion and adequate interpretation. Otherwise, the association of the two sets of data is not justified.
Response: Our focus was to investigate the combined effect on nutrients on GLP-1 secretion, which has not been shown before. We agree that our mRNA expressions data do not fully explain the signal transduction. Analysing the GPR120 and GPR40 receptor expressions reflects on the production of mRNA available which is in agreement with the previous publications and thus comparable (Kamakura R, et al. Mol Nutr Food Res. 2022 ;66(19):e2200192).
- Secretion of GLP-1 and expression of receptors, which are first to be clearly identified more than as GPR120, GPR40, etc., require a better understanding of the current status of knowledge as well as possible explanations of their relationships within the experimental system. Reasons for the choice of each analyzed mRNA should also be provided to explain the experimental logic and data interpretation.
Response: We used alpha-linolenic acid, which is a long-chain polyunsaturated fatty acid. It has been previously shown that receptors GPR120 and GPR40 are activated by medium and long-chain FFAs and GPR119 is activated by long-chain fatty acids (Talukdar S, et al. Trends Pharmacol Sci. 2011 ;32(9):543-50). There are several receptors for amino acid sensing in L-cells, but we measured CaSR, GPRC6A the key receptors for tryptophan and phenylalanine in L- cells (Modvig IM et al. Am J Physiol Endocrinol Metab. 2021;320(5):E874-E885). The discussion has been modified (Page 7 Line 207, 212, and 217): “αLA 6.25 µM and combination αLA 6.25 µM + Trp 5 mM has shown similar GPR40 mRNA expression which is mediated via fatty acid αLA. Our results showed an upregulating trend in CaSR expressions with L-Phe and L-Trp in STC-1 cells, however fatty acid αLA and control show similar CaSR expression. These indicate that the combination of αLA with amino acid (Trp/Phe) stimulates their respective receptor and potentiates GLP-1 secretion. In addition, we found that lower dose combination of αLA 6.25 µM + Trp 5 mM showed higher GPR120, GPR40, GPR119, and GPRC6A mRNA expression than higher dose αLA12.5 µM + Trp 10 mM. The reduced GLP-1 secretion with a higher dose combination of αLA with amino acids Trp/Phe might be due to desensitization of receptor signaling. The higher and lower dose combinations showed similar expressions of the above receptors”.
- Authors state that “monounsaturated FFA induced higher GLP-1 secretion than polyunsaturated or saturated FFA”. However, as polyunsaturated alpha-linolenic acid was used for their treatments, an explanation for this choice should be provided.
Response: The statement is now corrected (line 48 and 51; Page 2) “Unsaturated fatty acids are more potent GLP-1 secretagogues than saturated fatty acids in animals and humans (Thomsen C, et al. Am J Clin Nutr. 2003;77(3):605-11; Harden CJ, et al. J Proteomics. 2012;75(10):2916-23). Especially, long‐chain polyunsaturated fatty acids (LCPUFAs) such as docosahexaenoic acid (DHA 22:6, n‐3), α‐linolenic acid (αLA, C18:3, n‐3), and eicosapentaenoic acid (EPA, 20:5, n‐3) have shown a higher potency than other LCFA in suppression of body weight gain and appetite (Christiansen E, et al. Br J Nutr. 2015;113(11):1677-88; Kamakura R, et al. Mol Nutr Food Res. 2022 ;66(19):e2200192). Our main aim is to investigate combined nutrients increase GLP-1 secretions. For this we relied on our previous work (Kamakura R, et al. Eur J Pharm Biopharm. 2019 Nov;144:132-138; Kamakura R, et al. Mol Nutr Food Res. 2022 Feb;66(4):e2100978.
- Abstract reports that DPP4 inhibitor, still reported only as an undefined abbreviation, was used with nutrient combinations, but this is not described in results and methods. In addition, if DPP4 inhibitor was added to combinations, data should also include controls with it and without it.
Response: Thank you very much for the suggestion. Abbreviations were explained in abstract (line 21; Page 1) and described in results and methods (line 256 and 259; page 8): After 1 hour of preincubation, buffer was aspirated, and cells were treated with different nutrients and combinations of αLA, Phe, and Trp along with DPP4 inhibitor in KREBS buffer for 1 h at 37 °C: αLA 12.5 µM, 6.25 µM, Phe 6.25-25 mM and Trp 2.5-10 mM and their combinations αLA+Phe (i. 12.5 µM+ 25 mM, ii. 6.25 µM+ 12.5 mM, iii. 3.125 µM+6.25 mM) and αLA+Trp (i. 12.5 µM+ 10 mM, ii. 6.25 µM+ 5 mM, iii. 3.125 µM+2.5 mM) were used. KREBS buffer with ethanol (0.1%) and DPP4 inhibitor (0.25%) were used as control.
- Comments on the Quality of English Language: Minor English corrections are necessary as text in several points is not fully clear.
Response: The language has been corrected throughout manuscript.